# From “Maraschino” to Cell Cultures: A Deep Study on *Prunus cerasus* L. Cell Culture Juices

**DOI:** 10.3390/molecules30051089

**Published:** 2025-02-27

**Authors:** Vanessa Dalla Costa, Anna Piovan, Ina Varfaj, Maria Carla Marcotullio, Paola Brun, Raffaella Filippini

**Affiliations:** 1Department of Pharmaceutical and Pharmacological Sciences, University of Padova, Via Marzolo, 5, 35131 Padua, Italy; anna.piovan@unipd.it (A.P.); raffaella.filippini@unipd.it (R.F.); 2Department of Pharmaceutical Sciences, University of Perugia, Via Fabretti 48, 06123 Perugia, Italy; ina.varfaj@dottorandi.unipg.it (I.V.); mariacarla.marcotullio@unipg.it (M.C.M.); 3Department of Molecular Medicine, University of Padova, Via Gabelli 63, 35121 Padova, Italy; paola.brun.1@unipd.it

**Keywords:** *Prunus cerasus*, plant calli, suspension cultures, naringenin, anti-inflammatory activity, antioxidant activity

## Abstract

*Prunus cerasus* var. Marasca (Rosaceae) is an important Croatian cultivar, known wordwide for the production of Luxardo maraschino liqueur, which occurs in the eastern Po Valley of Italy. Besides liqueur, Marasca is attractive for its beneficial effects on human health and well-being. The undifferentiated in vitro cell cultures of Marasca were investigated as a source of healthy products. The in vitro conditions for obtaining callus and suspension cultures under photoperiod were defined. The cell lines were evaluated for growth rate, total phenol and proanthocyanidin contents, and antioxidant activities via colorimetric assays. The best cell lines were also subcultured in darkness, studying the importance of the light parameter in the possible industrial scaling-up. The juices extracted from the obtained biomasses were analyzed by LC-DAD-MS and six flavanone derivatives, among which naringenin and its glucoside were identified. The quantitative analysis, pursued during the cell growth cycle, revealed that the flavanone content was higher at the end of the growth cycle (28th day) and that the total content of identified flavanone compounds varied from 17.22 to 79.22 μg/mL of juice. The results of the antioxidant and anti-inflammatory activities on Caco-2 cells support the potential applications of this material in human health.

## 1. Introduction

The *Prunus* genus (Rosaceae) is characterized by several species and varietal biodiversity that are found in wild, semi-wild, and cultivated forms worldwide. Although more than thirty cherry species, mostly indigenous to Europe and Asia, have been identified to date, only fruits grown mainly from two *Prunus* species are globally traded, commonly called sweet cherry (*P. avium* L.) and sour or tart cherry (*P. cerasus* L.) [1].

*P. cerasus* originated around the Caspian Sea and is presumably native to northwest and central Europe. Recent molecular and genetic analysis supports the hypothesis that sour cherry represents an inter-specific hybrid between ground cherry (*P. fruticosa* Pall.) and sweet cherry. The most widely cultivated varieties are ‘Montmorency’ in the United States, ‘Kütahya’ in Turkey, and ‘Marasca’ and ‘Balaton’ in Europe [2]. *Prunus* species have various characteristic features, including leaf glands; an upper ovary; a single carpel, with each carpel containing two drooping artificial ovules; and stone fruit that has a juicy mesocarp, which can be fleshy or dry and does not divide when mature, or splits on its own in rare cases [1].

Sour cherry var. Marasca (*P. cerasus* var. Marasca) is an important and well-known Croatian cultivar. Marasca is a deciduous tree with small leaves; the fruits are dark red, and their size and chemical composition can change based on climatic and agricultural conditions [3,4].

The low amount of sugars and high content of organic acids contribute to the low palatability of sour cherries, leading to their use in processed materials such as juice, jam, frozen or dried fruits, and liquors, rather than fresh ones [5,6]. Despite its less appreciated organoleptic properties, sour cherry represents a concentrate of nutritive (simple sugars and small quantities of proteins) and non-nutritive compounds (vitamins A, B1, B2, B3, C; minerals K, Fe, Ca, P), together with polyphenols, anthocyanins, and flavonols, which are highly correlated with antioxidant, antiradical, and disease preventing potential [5,6,7,8]. In traditional medicine, the fruit of sour cherry is also used as prophylactic to avoid cardiovascular damage, Alzheimer’s disease, inflammatory diseases, and chronic ailments marked by elevated oxidative stress, such as cancer and diabetes. In addition, cherries improve appetite, lower blood pressure, protect against oxidative stress, reduce pain and muscle damage caused by exercise, and modulate blood glucose, and are helpful as a treatment for insomnia [9,10,11,12,13,14,15,16]. Beyond the fruit, other plant parts can also be considered a promising source of phenolic compounds. Even if a complete phytochemical evaluation is, as far as we know, not reported for the Marasca variety, a recent work reported the composition of the phenolic compounds in leaf samples of *P. cerasus* cultivars, with total contents ranging from 8.254 to 16.199 mg/g in dry leaves [17].

In Italy, *P. cerasus* var. Marasca is cultivated in the eastern part of the Po Valley, in the Euganean Hills, where the soil made of volcanic, calcareous, and sedimentary limestone is a favorable Mediterranean area for the cultivation of this plant. Properly in this region, one of the most prominent Italian cultivations of Marasca is pursued by the Luxardo company (https://www.luxardo.it/it/la-marasca-luxardo; accessed on 22 November 2024). Luxardo is one of the most famous producers of products derived from this cherry, particularly the well-known Luxardo Maraschino Liqueur.

Due to the increased impact of Marasca upon the fruit market and its great potential in different fields, in our study, we decided to evaluate this important regional cultivation, using in vivo plant leaves of Luxardo Marasca as starting material to obtain undifferentiated in vitro cell cultures. In vitro plant cell culture is the general term for the culture of cells, tissues, organs, and plants on synthetic media under aseptic conditions and the definition of physical and chemical parameters, the so-called in vitro conditions. Plant cell cultures have shown great potential for food purposes, and this technology can constitute an additional tool for global food production and, possibly, for the nutraceutical industry, being an alternative system for the sustainable production of metabolites of pharmacological, cosmetic, and food interest [18,19,20,21,22].

The literature related to the establishment of *Prunus cerasus* in vitro cultures is scarce and, in part, focused on micropropagation processes, e.g., [23,24], and none of these is related to the Marasca cultivar. In fact, one of the first protoplast-derived calli of *P*. *cerasus* was gained in the 1990s [25] as part of the method of the plant regeneration process.

As far as we know, only two published papers involve callus obtainment, and they are exclusively focused on in vitro anthocyanin production. Since anthocyanins obtained from fruit depend on seasonal availability and, moreover, since the fruit has too high a value as a fresh fruit to be used for anthocyanin extraction, the authors investigated plant cell cultures as a viable opportunity for continuous production of this class of secondary metabolites. In the first work [26], the authors compared anthocyanidin profiles in calli and fruits, highlighting differences in qualitative and quantitative composition. In a subsequent publication, the authors evaluated the anthocyanin production of calli under different growing conditions. Regarding anthocyanin content, the authors did not find significant differences in the leaves and calli [27].

Since several plant cell cultures have been directly exploited for the commercial production of phytochemicals to be used as pharmaceuticals, pigments, and ingredients for cosmetics and food [28], this work aimed to obtain undifferentiated in vitro cell cultures of *P. cerasus* var. Marasca, which, to date, have not been investigated. The biosynthetic potential and possible antioxidant and anti-inflammatory activity on Caco-2 cells were investigated on the biomasses obtained. The potential positive outcomes of this research can highlight plant cell cultures’ potential to be utilized for the production of innovative healthy products and foodstuff, contributing to the development of a more sustainable supply.

## 2. Results and Discussion

### 2.1. In Vitro Culture Establishment, Preliminary Colorimetric Assays, and Cell Line Selection

#### 2.1.1. Callus Culture Establishment

In this study, we assessed the potential of in vitro cultures of *Prunus cerasus* var. Marasca, using, as an explant source, in vivo plant leaves from an Italian sour cherry cultivation, prompted by the positive outcomes of plant cultures in increasing the value of plant-derived materials.

Even if the methodology for initiating the in vitro cultures is well established nowadays, it remains tricky. In fact, different parameters must be considered case by case, e.g., explant type, explant sterilization, and nutritional medium composition [29,30]. Sterilization was the first parameter considered since it is the essential prerequisite for introducing any plant material into in vitro conditions. The process must be balanced between the decontamination’s success and the material’s integrity. In general, the nature of the material influences the contact time with the sterilization agent and its concentration. No correlation was observed between the contact time with the sterilization agents and the contamination. Probably due to the harvesting season (autumn), the explants ended in high contamination, and the overall percentage was between 50 and 80%.

On the survival explants, after the sterilization step, a brown color was observed on all the explant surfaces. The browning in plant tissue culture refers to a phenomenon in which the explants release brown substances or phenolics during de-differentiation and/or re-differentiation; these compounds are secreted from wounded regions of explants as a defense response, and oxidation of these compounds results in browning of culture media and plant tissues [31].

Regarding callus induction, the choice of the basal medium as well as plant growth regulators was mainly based on the empiric experience and, only marginally, on literature data. There are few published papers related to the establishment of in vitro cultures of *Prunus cerasus*, and none of these is focused on the cultivar Marasca. In three studies, calli were obtained using an MS [32] basal medium, and different hormonal balances were obtained using cytokinins and auxins [25,26,27].

In our experiments, we decided to test MS [32] and B5 [33] basal media, which are some of the most used, together with WPM [34], which is used for woody plants, as in our case [35]. Likewise, we tested a combination of auxin and cytokinin in a 1:1 ratio. To test the effect on the callogenesis, we separately used two different auxins: 2,4-dichlorophenoxyacetic acid (2,4-D) and naphtalenacetic acid (NAA), with the same cytokinin, kinetin (K), added.

The callogenesis started in all media, except MS NAK (MS basal medium plus NAA and K) and MS DK (MS basal medium plus 2,4-D and K), despite the explant browning. Calli formed in correspondence to the area where the leaf was cut to obtain the explants. Figure 1 shows leaf explants with pale green-colored and soft-textured calli protruding from the mesophyll.

Table 1 shows the percentage of callogenesis and the callus size after three subcultures in each medium.

As reported in Table 1, as compared with other media, calli in WPM NAK (WPM basal medium plus NAA and K) exhibited the highest callogenesis percentage, but they kept small dimensions, whereas calli in WPM DK (WPM basal medium plus 2,4-D and K) continued the growth, almost reaching the growth rate of calli in B5 NAK (B5 basal medium plus NAA and K) and overtaking the dimensions of calli in B5 DK (B5 basal medium plus 2,4-D and K). Calli in B5 NAK were the ones with the highest growth rate. In addition, in the following months, calli in WPM (especially WPM DK) tended to shrivel and reduce their growth rate compared to calli in B5; therefore, we concluded that B5 NAK and B5 DK media were the better choice for the percentage of callogenesis, callus size, and callus maintenance.

Calli in both media (B5 NAK and B5 DK) showed the co-existence of more than one cell line within the same medium. At each subculture, the cell lines were separated to stabilize them, and only after two years of subcultures did they finally stabilize into two distinct cell lines on both the selected media. One cell line was bright green in B5 NAK and pale green in B5 DK, while others appeared brownish in both B5 NAK and B5 DK (Figure 2).

The two green cell lines, P NAK V (in B5 BAK medium) and P DK V (in B5 DK medium) were characterized by more-or-less friable and juicy callus biomasses, whereas the brown ones of P NAK M (in B5 NAK medium) and P DK M (in B5 DK medium) were constituted of harsh and nonhomogeneous material. During the subsequent subcultures, the growth rate of the brown cell lines was remarkably lower than that of the green ones. In fact, starting from the same quantity of calli at the time of transfer to the new culture medium, less than half of the cell material was obtained compared to the green lines, as can also be observed in Figure 2 (picture taken for all the cell lines at the end of growth cycle).

#### 2.1.2. Preliminary Colorimetric Assays and Cell Line Selection

After about 8–12 subcultures, preliminary colorimetric assays were carried out on each cell line (P NAV, P DK V, P NAK M, and P DK M) to evaluate probable differences in the secondary metabolism of morphologically different cells, as well as to deny or confirm the similarity between the material obtained in the two media. The colorimetric analyses were also performed on brown cell lines, for which the growth rate resulted in a limited factor, in order to, in this case, definitely discard these materials.

Considering that in vitro cell metabolism can change during the growth cycle as a result of the cellular activity, these assays were followed on juices of material harvested on the 14th, 28th, and 42nd day of the growth cycle (the period elapsing between two subcultures before the necessity of transferring it).

We have chosen not to use solvent extraction mainly for two reasons. The first one is to reduce the possibility of artefact formation. It is well known that solvents may interact with various analytes through chemical reactions. The second reason concerns the toxicity of the solvent and ecological damage, also in view of the potential use of in vitro plant cell materials in food-related industries [36,37].

Due to their importance in plant life and human health, the total phenol and the total proanthocyanidin contents (TPC and TPcC, respectively) were assessed. Polyphenols are the most prevalent class of secondary metabolites found in plants, playing a critical role in their chemical defense mechanisms against biotic and abiotic stresses. These compounds are recognized for their potent antioxidant properties, which can help to prevent various severe health conditions in humans. The determination of the total polyphenol content in plant material is a critical step for understanding biological effects, such as the antiradical, antioxidant, and anti-inflammatory properties. Numerous studies have indicated that the elevated levels of anthocyanins in *P. cerasus* var. Marasca, and other phenolic compounds present in the fruit, determine significant antioxidant and anti-inflammatory activities that positively influence human health [5,6,8].

Although many plant bioactive compounds have been extensively investigated in the past, others are still partially unknown. The proanthocyanidin class is among these latter. In plants, their main role is represented by the first biochemical defense against external injuries. The overproduction of reactive oxygen species in response to environmental stresses leads to the activation of specific metabolic pathways for the production, among the other defence mechanisms, of antioxidant molecules. It is known that proanthocyanidins are the result of the condensation of at least two flavan-3-ol units, and this structural complexity increases the antioxidant properties of these compounds [38]. For the reasons above, proanthocyanidins are a class of polyphenolic compounds that are receiving significant interest in the nutraceutical field.

Ultimately, given the correlation observed between the classes of secondary metabolites analyzed and their attributed antioxidant activity, the antioxidant activity was evaluated in the juices. This assessment aimed to establish a relationship between secondary metabolite content and their corresponding antioxidant activity.

The TPCs, expressed as µg of gallic acid equivalent/mL of juice, of P NAK V, P DK V, P NAK M, and P DK M juices are illustrated in Figure 3.

Looking at the results, it is evident that the green cell lines, PNAK V and PDK V, displayed the highest phenol contents *(p* < 0.001) at each time point compared to the brown cell lines (P NAK M and P DK M); moreover, they exhibited the same statistically greatest content among the samples at 14 days, with 234 and 239 μg of phenol content per mL of juice, respectively. On the other hand, the two brown cell lines, PNAK M and PDK M, reached their maximum at 28 days, with the same statistical phenol content, 113 and 112 μg/mL of juice, respectively. In each cell line, the TPC results were lower on the 42nd day and were significantly different from the other time points, except for P DK V, which resulted in the second highest one (212 μg/mL of juice).

The TPC analysis revealed differences mainly between the two different color cell lines, with the green cell lines being the richest ones at each time point. In contrast, the TPcC analysis reported that both at 14 and at 28 days, P DK M juice is one of the richest in catechin expressed as µg of equivalent/mL of juice, its content being statistically comparable with P NAK V at 14 days. All three conditions reported values of catechin above 50 μg/mL of juice. Looking at the single cell line, the content of catechin reduced in the juices at 42 days compared to 14 and 28 days, the values reported in P NAK V, P DK V, and P DK M were statistically similar (*p* > 0.05), whereas P NAK M reported the lowest value (33 μg/mL, Figure 4).

Figure 5 shows the results of the antioxidant activity. The correlation between the juice volume and the activity was confirmed; in fact, the activities (expressed as percentage of DPPH inhibition) resulted in about a two-fold increase when 50 μL were tested, compared to 25 μL, except for P NAK M on the 42nd day and P DK M on the 14th and 42nd days. As for the TPC, the results of the antioxidant activity demonstrated that the two cell lines with similar morphological characteristics show a comparable activity trend. PNAK V and PDK V exhibited the highest activity, whereas both the brown cell lines showed low antioxidant activity (*p* < 0.0001). The antioxidant activity, in the same reaction condition, of 50 μL of the poorest juice, P NAK M, on the 42nd day, and the richest, P DK V on the 42nd day, are comparable to 0.25 and 3 μg of ascorbic acid, respectively.

As for the TPC, the results of the antioxidant activity also demonstrate that the two cell lines with similar morphological characteristics demonstrate a comparable activity trend. The measured antioxidant activity seems to be mainly attributable to the phenolic compounds present in the juices, confirming a good correlation between the phenolic content and antioxidant capacity, as it is often the case [39].

Considering the scarce industrial applicability of the brown lines, together with a low secondary metabolism, the material was discarded, and the further experiments were conducted only on the green lines P NAK V and P DK V.

In order to evaluate the influence of the light on the growth and the biosynthetic potential, some of the P NAK V and P DK V calli were also subcultured in dark conditions (P NAK Vd and P DK Vd, hereafter).

Light is an important environmental factor affecting plant growth, development, and metabolism; physiological and biochemical reactions are triggered by changes in light quality, intensity, and duration [40]. For in vivo plants, light is an important physical factor for plant cell cultures, as it affects in vitro morphogenetic responses of cultured cells and tissues [41]. The photoperiod influences tissue cultures, as it remarkably promotes the cell growth, morphogenesis, and accumulation of secondary compounds [40]. For these reasons, our plant cultures are grown under a 16 h light and 8 h dark photoperiod. However, the growing of cultures under illuminated conditions requires more technical efforts and energy, being, therefore, costly [42]. After a period of subculture in darkness, the biomasses were observed during the growth cycle to highlight possible morphological differences between the material grown under photoperiod and dark conditions.

Visual observations of the material were done both during the assessment period in the dark conditions and in the following subcultures. In the dark, calli of P NAK Vd changed color, from bright green to pale brown. The texture of the calli did not change, even if they slowed down the growth rate. Calli of P DK Vd showed quite the same response to darkness, changing the color from bright green to pale brown; their growth rate was slower than that of calli under photoperiod and also slower than that of P NAK V calli under dark conditions. Figure 6 shows the calli of P NAK Vd and P DK Vd.

### 2.2. Suspension Cultures

In the context of a scaling-up to the industrial level with bioreactors, which rarely allow cell exposition to light [43], in this study, we investigated whether *P. cerasus* cell suspension cultures could be cultivated in dark conditions. Suspensions from both P NAK V and P DK V were established. After the inoculation, P NAK V showed a high proliferation, reaching the appropriate density fast, and the cells kept the original calli colour, forming greenish suspensions. On the other hand, P DK V suspensions were characterized by a slower growth compared to P NAK V, and the cell tended to oxidize. Beyond the growth parameter, the decision to continue only with the P NAK V cell line was also made based on the abundance of P NAK V biomasses, together with the absence of 2,4-D as phytohormone in the culture medium. Therefore, as was done for calli, we checked whether our suspensions could be cultivated in dark conditions while maintaining the same morphological characteristics and secondary metabolism. Figure 7 shows P NAK V suspension in darkness (P NAK Vs, hereafter), and, on the right, a magnification of the cell suspension. The observation under the light microscope showed the presence of single spherical cells and very small aggregates, an important characteristic to obtain a homogeneous suspension and a suitable mass transfer.

Measuring suspension growth parameters is important for reproducible batches both at the laboratory and at the industrial levels; therefore, the suspension growth was measured during the time frame of 31 days. The cell volume after sedimentation (CVS) method was used to measure the volume of a single flask without killing the cell. The CVS was measured every two to three days; Figure 8 shows the growth curve of P NAK Vs. The results are expressed as cell volume mean ± standard deviation during a timeframe of 31 days. The growth curve showed the typical shape, with a 6-day logarithmic phase, an exponential growth phase between the 7th and 29th days, and the start of the stationary phase on day 32.

### 2.3. Phytochemical Analyses

#### 2.3.1. LC-MS and HPLC-DAD Qualitative Analysis

The metabolic composition of the juices was investigated by LC--MS/MS and HPLC-DAD. The tentative identification was pursued on calli under photoperiod, P NAK V and P DK V, on calli in dark conditions, P NAK Vd and P DK Vd, and on the cell suspension in dark conditions, P NAK Vs.

The retention times, the maxima of UV absorptions, the parental ions expressed as [M-H]^−^ *m*/*z*, the fragmentations and the tentative compound characterizations are reported in Table 2. All the reported data were manually observed, and the compound presence or absence in the four cell lines is marked with “x” or “-” respectively.

The studies on *P. cerasus* cell cultures are mainly focused on anthocyanin biosynthesis because of the problems associated with anthocyanins obtained from the fruit [26,27]. In our study, we did not detect this class of flavonoids, demonstrating that the entire biosynthetic pathway leading to their formation is not active. We mainly focused on another subclass of flavonoids, the flavanones, which represent the first products of the flavonoid biosynthetic pathway and are interesting therapeutic agents [44].

A total of six flavanones were detected. Compound **6** was identified as naringenin by comparison with the reference standard; by considering the parental ions and fragmentation patterns, compounds **4** and **5** were inferred as naringenin derivatives. Compound **5** resulted as a naringenin glucoside, based on the typical glucose unit losses of a hexose moiety (−162 Da). We could not entirely characterize compound **4**, for which we recognized the daughter ions of 271 Da, corresponding to naringenin. Compounds **1** to **3** were catalogued as flavanone derivatives based on the characteristic flavanone UV spectra, with the absorption peak in the range 270–295 [45]. Moreover, these latter compounds exhibit the same parental ions (*m*/*z* 433), but different fragmentation patterns, probably due to the different position of some functional groups in the backbone moieties. In support of this, it was demonstrated that multiple fragmentation routes exist from a single precursor ion (e.g., glucose), classified in major and minor pathways [46].

Figure 9 shows P NAK V chromatogram (as an example), acquired at 325 nm, with the identified peaks numbered.

In the calli cultured under photoperiod, all six compounds were found in P NAK V juice, while only four compounds were found in P DK V. Meanwhile, on the calli in dark conditions, six compounds were found in P DK Vd and five in P NAK Vd. Notably, cultured in dark conditions, the suspension P NAK Vs resulted in qualitative overlapping to P NAK V calli under photoperiod.

#### 2.3.2. Quantitative Analysis

Quantitative analyses were performed on P NAK V, P DK V, P NAK Vd, and P DK Vd juices obtained from calli harvested on the 14th and 28th days of the growth cycle, and on P NAK Vs juice from suspension’s cells harvested on the 15th.

Considering the general decrease in P NAK V and P DK V juices in secondary metabolite contents at 42 days (except for P DK V TPC), the quantitative analyses refer only to 14 and 28-day-old callus juices. Similarly, the suspension’s cells were harvested and the juice obtained during the logarithmic phase (at 15 days), since the production of secondary metabolites more often occurs near the stationary phase [47], but also during different moments of the lag phase [48]. The choice of using mid-growth cycle material was based on the preliminary callus analyses, in which the phenolic content was shown to decrease at the growth cycle end.

The peak areas of the six identified compounds were plotted against the naringenin standard curve (acquired at 325 nm), and the resulting concentrations were summed and expressed as total flavanone content for each cell lines. The total flavanone contents, expressed as µg/mL of juice for every cell line, are reported in Table 3.

P NAK Vd juice at 28 days turned out to be the richest sample, with a total flavanone content of 79.22 µg/mL of juice, statistically different from all the other samples (*p* < 0.0001). Moreover, both P NAK V and P NAK Vd juices exhibited statistically higher (*p* < 0.001) flavanone contents at 28 days of growth, compared to 14 days. The same did not happen for P DK V and P DK Vd juices, in which the contents were the same at the two time points for P DK and, even more, also statistically lower for P DK Vd at 28 days than at 14 days. The suspension P NAK Vs, even though qualitatively comparable to the parental calli P NAK V, exhibited a significantly poorer total flavanone content compared to it (*p* < 0.0001), and comparable to P NAK Vd at 14 days of the growth cycle. The P DK V juices resulted in the poorest samples at both the times.

Light is an important physical factor for plant cell cultures, affecting in vitro morphogenetic responses [41]. Moreover, not only the light quality and intensity but also the photoperiod is a crucial environmental factor; tissue cultures are influenced by the photoperiod, as it remarkably affects the cell growth, morphogenesis, and accumulation of secondary compounds [40]. Our results support the possibility of suspension obtainment in dark conditions, and also highlighted that for all the biomasses analyzed, the total flavanone contents were higher in the dark with respect to the material grown under photoperiod.

Besides the total flavanone contents, in Figure 10, the amounts of each compound in the juices are reported. As shown in Figure 10, compound **5** (naringenin glucoside) is the prevalent compound in all the juices, and in P NAK Vd, it reached 43.86 µg/mL on the 28th day, almost fifty per cent of the total flavanone content. Meanwhile, the other naringenin derivative (compound **4**), instead, was found only in P NAK V and P DK Vd calli (on the 14th and 28th days of growth, respectively) and in P NAK Vs suspension in a greater amount. Moreover, in the suspension juice (P NAK Vs), one of the highest naringenin contents, 6.05 µg/mL of juice, was found, second only to P DK Vd naringenin content (7.58 µg/mL).

To the best of our knowledge, naringenin and naringenin derivatives are not among the most important constituents of *P. cerasus*. The presence of naringenin was reported in heartwood, along with flavone and flavonol compounds [49], and in kernel oil (1678 µg/kg) of *P. cerasus* [50]. Akšić et al. [51] detected naringenin in Oblačinska’ sour cherry (*Prunus cerasus*) pollen, with content within a range of 0.04 to 0.18 mg/kg in different clones. *Prunus* aside, it is recognized that naringenin derivatives are characteristic metabolites of the *Citrus* genus, in which they are present in high quantities [52]. Since high levels of these compounds are associated with plants’ young developing states, they can be found mainly in immature tissue. Therefore, plant cell cultures can be used as an alternative source. Properly related to this possibility, some authors studied the content of two compounds, of which one was naringin from *Citrus aurantium* cv. Sevillano calli. They found naringin present at 9 mg/g in dry callus, which is around nine times the total flavanone content in our dry richest sample, P NAK Vd (data not shown). Moreover, the authors confirmed the greater bio-productive capacity of younger tissue; in fact, they found more naringin in the external zone of the callus [53]. The results of our study support the great potentiality of Marasca in vitro cell cultures to be used as an alternative source of flavanone derivatives, even if we did not use the *Citrus* genus as a callus source and the extraction was not selective for this compound class.

Since secondary metabolism modification is one of the major issues related to the establishment of the suspension, the most interesting result of our study is that the suspension juice not only maintained the same phytochemical profile but also quite the same qualitative ratio. Cell suspensions with a good growth rate and an active secondary metabolism, like the one we obtained, are the starting conditions to obtain material that deserves to be valuable for the scaling-up.

### 2.4. Activity Assays

#### 2.4.1. Identification of Non-Toxic Concentrations of Juices

The effect of P 1 (*Prunus* callus P NAK V, under photoperiod), P 2 (*Prunus* callus P NAK Vd, in the dark), and P 3 (*Prunus* suspension P NAK Vs, in the dark) juices have been tested on Caco-2 cells, a cell line helpful in studying, selecting, and classifying drug candidates under controlled conditions [54].

Before the activity assays, the non-toxic concentrations of juices were assessed. Caco-2 cells were incubated with the juices at concentrations ranging from 0 to 50% vol/vol with 1:2 serial dilutions. Cell toxic effects were evaluated using the MTT test.

Table 4 reports the highest non-toxic concentrations determined in our experiments. Among the tested extracts, P1 and P2 reported the lowest toxic effects; indeed, 25% vol/vol demonstrated cell viability comparable with non-treated cells. Moreover, we observed that P3 reported the highest toxic effects compared to P1 and P2. To compare the extracts, we selected the lowest non-toxic concentrations (used concentrations) in the subsequent experiments, as reported in Table 4.

#### 2.4.2. Intracellular ROS Reduction

It is proven that free radical generation is associated with the initiation of human diseases. The imbalance between the synthesis and accumulation of reactive oxygen species (ROS) in cells and tissues and the capacity of an organism to eliminate these reactive compounds results in oxidative stress. Otherwise, the maintenance of a healthy life depends on the supply of antioxidant nutrients from the diet [55].

A published paper [56] demonstrated the effectiveness of a polyphenol-rich extract of *Prunus avium* fruits in reducing intracellular ROS production in Caco-2 cells. Because of the abundance of polyphenols in the material obtained, and since polyphenols have a relatively short half-life [57], to evaluate the potential antioxidant effects of the juices, we pre-treated Caco-2 cells with extracts twice, and then we challenged the cells with H_2_O_2_ to induce intracellular ROS burst. As reported in Figure 11, H_2_O_2_ increased by 5.8-fold ROS production in cells (*p* < 0.0001) vs. non-treated cells. *Prunus* materials (P 1, P 2, and P 3) were effective (*p* < 0.0001) in reducing ROS levels in H_2_O_2_-challenged cells, especially callus grown in the dark (P 2) and suspension juices (P 3). These results contrast with data reported in Table 3, where P NAK V suspension (P3) reported a significant reduction in total flavanone content compared with the parenteral calli. Despite the differences in flavanone contents, we hypothesise that the flavanone concentration in P3 is sufficient to ensure the antioxidant effects in Caco-2 cells. Moreover, the similar antioxidant activities in P2 and P3 could be explained by (i) the different bioavailability of antioxidant compounds, increasing the cellular uptake, or (ii) the synergistic interaction of flavanone with other components, triggering different antioxidant pathways in the cells.

#### 2.4.3. Effects on Lipopolysaccharide-Induced Inflammation

During the oxidative stress process, the reactive oxygen/nitrogen species can initiate intracellular signalling through which specific pro-inflammatory pathways are expressed. On the other hand, oxidative stress can be enhanced by inflammation. Inflammation represents a pathophysiological mechanism of defense that acts in case of homeostasis perturbations; however, an uncontrolled inflammation may be potentially harmful and lead to many acute and chronic diseases [58]. Inflammation is triggered by a wide range of external (microbial) and internal factors (due to cell lysis). The initial stages of the inflammatory response include the activation of macrophages and the secretion of, among others, cytokines like interleukin-1β (IL-1β) and tumor necrosis factor α (TNF-α), which are pro-inflammatory factors [37]. Thus, we tested the anti-inflammatory activity of the juices by evaluating the reduction in the production of IL-1β and TNF-α in Caco-2 cells challenged with bacterial lipopolysaccharide (LPS).

The concentrations of juices used for the anti-inflammatory assays are reported in Table 4, and are the same as for the ROS assay. Caco-2 cells were incubated for 16 h with juices alone. The pro-inflammatory cytokines’ levels were assessed in the conditioned media by ELISA. We observed (Figure 12A,B) that extracts per se have no pro-inflammatory effects in Caco-2 cells; indeed, levels of the pro-inflammatory cytokines IL-1β and TNF-α did not significantly increase in cells incubated with juices compared to non-treated cells. Stimulation with LPS enhanced the production of IL-1β and TNF-α compared with non-treated cells (Figure 12C,D, respectively; *p* < 0.0001). All the tested juices reduced the production of IL-1β and TNF-α in LPS-stimulated cells (*p* < 0.0001, Figure 12). We did not detect significant differences among calli under photoperiod, calli in the dark, and suspensions maintained in dark conditions, meaning that all the biomasses preserved their anti-inflammatory capabilities.

## 3. Materials and Methods

### 3.1. Plant Material

#### 3.1.1. Callus Cell Cultures

Leaves from in vivo-grown plants were harvested from *Prunus cerasus* var. Maraska Luxardo growing on Euganean Hill during the last week of September 2022 and used as starting material.

Plant material was washed accurately with tap water with a detergent added. The sterilization process was carried out by dipping the plant material in 80% ethanol for 60 sec., and in 14% sodium hypochlorite plus a wetting agent (Teepol) for 11 and 15 min, to evaluate the right balance between the sterilization process and survival of explants. After the sterilizing agent removal with sterile distilled water (three times), the leaf blades were cut into an area of almost 1 cm^2^, and the primary explants were introduced into the culture media.

For callus induction, three basal media were used: MS [32], B5 [33], and WPM [34]. All the media were supplemented with sucrose (30 g/L) as a carbon source and plant hormones in different balances (Table 5). The media was solidified with agar (10 g/L); the pH was adjusted to 5.7 before autoclaving.

All the experiments were performed in Petri dishes using 10 explants per dish. The explants were cultivated in a growth chamber at 25 ± 1 °C in a 16 h photoperiod, and the obtained biomasses were subcultured every 3–6 weeks. Responses were recorded weekly. Cells were regularly observed with the naked eye and under stereomicroscope (S9i-Leica).

To examine the effect of light on callus growth and composition, stabilized callus cultures (after 12 subcultures) were partly maintained in photoperiod (16/8 h) and partly in the dark.

#### 3.1.2. Suspension Cell Cultures

Cell suspension cultures were initiated from established P NAK V and P DK V strains grown in photoperiod, by transferring a suitable quantity of friable calli (1–3%) in 500 mL Erlenmeyer flasks containing 100 mL of liquid B5 NAK and B5 DK media, respectively. The pH was adjusted to pH 5.7. The cultures were maintained on an orbital shaker at 100 rpm at 25 °C under photoperiod conditions and subcultured every 14–32 days. After 4 subcultures, the stabilized suspensions were transferred in dark conditions.

The growth curves of cell suspension cultures were calculated based on the Cell Volume after Sedimentation (CVS). The measurement was taken using a device that guarantees a fixed position of the flask, with an angle of 60°, and a ruler fixed onto the device which enables the reading of the height of the sedimented cell mass. The height of the cell mass, after 7 min of sedimentation, represented the volume of the cells, previously calibrated by measuring different known volumes of water [59]. Cells were regularly observed under optical microscope (Inverted light microscope—TELAVAL 31—Carl Zeiss).

### 3.2. Chemical Analysis

#### 3.2.1. Juice Preparation

Juices from calli and cell suspensions were prepared. Calli were taken from Petri dishes and maintained at −18 °C. The cells from suspension cultures were filtered and washed with distilled water; then the suspension cells were left to dry until they lost the surrounding water, and maintained at −18 °C. The materials were thawed and squeezed (adding quartz powder) with a pestle. The ground samples were put in an ultrasound bath for 40 min. After centrifugation at 12,000 rpm, the supernatant of each sample (juice, hereafter) was taken and analyzed.

#### 3.2.2. Colorimetric Assays

Total phenol content. The total phenol content was determined using the Folin–Ciocâlteau colorimetric assay as described by [60], and gallic acid was used for the standard calibration curve. A quantity of 200 µL of undiluted or appropriate diluted juice was added to 1 mL of 1:10 diluted Folin–Ciocalteu reagent. After 4 min, 800 µL of sodium carbonate (75 g/L) were added. After 30 min of incubation at room temperature, the absorbance at 765 nm was measured using a HeλIOS (Thermo Electron Corporation; Rockville, MD, USA) spectrophotometer. The results were expressed as gallic acid equivalent μg/mL of juice. The calibration curve was constructed: y = 0.0424x + 0.0106 R^2^ = 0.9997 (gallic acid 1–20 µg/mL). All the samples were analyzed in duplicate and analyzed twice a week apart; the results were reported as means ± standard deviation (SD).

Total proanthocyanidin content. The total proanthocyanidin content was quantified using the vanillin colorimetric assay, with catechin used to create the standard calibration curve, according to modified [61]. A quantity of 25 µL of undiluted or appropriately diluted samples were mixed with 1 mL vanillin solution at 1% in H_2_SO_4_ at 70%. Following a 15-min incubation at room temperature, the absorbance of the samples and the standard was read at 500 nm. All the samples were analyzed in duplicate, and the results were expressed as the mean ± standard deviation (μg/mL) of juice. The calibration was constructed: y = 0.1233x−0.0022 R^2^ = 0.9998 (catechin 0.2–2.5 μg/mL). All the samples were analyzed in duplicate and analyzed twice, a week apart; the results were reported as means ± standard deviation (SD).

Antioxidant activity. The antioxidant activity was tested using the DPPH method assay [62]. A quantity of 100 µL of undiluted or appropriately diluted sample or of ascorbic acid, used as standard, were mixed with 400 µL DPPH solution (0.1 mM), and after 30 min of incubation, the absorbance at 517 nm was read. The antioxidant activity was calculated as the percentage of inhibition based on the discoloration of DPPH solution in the presence of antioxidant activity. The juices of each sample were tested, and for each time point, two different volumes of juice were tested, in order to assess a volume–activity correlation. All the samples were analyzed in duplicate and analyzed twice, a week apart; the results were reported as means ± standard deviation (SD).

#### 3.2.3. HPLC-DAD Analysis

HPLC-DAD analysis was performed using an Agilent 1100 HPLC Series System (Agilent, Santa Clara, CA, USA) equipped with a degasser, quaternary gradient pump, column thermostat, and UV-Vis detector. A Gemini 5 µm C6-Phenyl column (250 × 4.6 mm) from Phenomenex (Torrance, CA, USA) was employed at 40 °C. The mobile phase consisted of 0.15% acetic acid in water (A) and acetonitrile (B), with the following gradient elution program: 97% A at 0–6 min, 75% A at 15 min, 75% A at 20 min, 20% A at 30 min, and 97% A at 40 min. The flow rate was 1 mL/min, with an injection volume of 10 µL; chromatograms were acquired at 265 and 325 nm, and UV–Vis spectra were recorded in the 190–700 nm range [37].

For the quantification of naringenin derivatives, the chromatograms acquired at the wavelength of 325 nm were used. The content was expressed as naringenin derivatives using authentic commercial standard. Naringenin standard solution (1 mg/mL) was prepared in methanol, and the calibration curve was obtained in a concentration range of 4–80 µg/mL, with six concentration levels. Peak areas were plotted against corresponding concentrations (R^2^ = 0.9995). The analysis was performed in triplicate, and the results were expressed as mean ± standard deviation (SD).

#### 3.2.4. LC-MS/MS Analysis

The analysis was performed using an Agilent 1290 Infinity II combined with the Agilent 6550 mass spectrometer (Agilent, Santa Clara, CA, USA). The chromatographic separation was done by a Gemini 5 μm C6-Phenyl column (250 × 4.6 mm) from Phenomenex (Torrance, CA, USA). The optimized gradient program was the following: eluent A water with 0.1% (*v/v*) formic acid, eluent B acetonitrile; 0–8 min, 97% A; 8–26.5 min, 75% A; 26.5–40 min, 20% A; 40–42 min, 97% A. The flow rate was 0.75 mL/min, with an injection volume of 5 μL; the column temperature was kept at 40 °C. Chromatograms were acquired at 265 and 325 nm; and UV–Vis spectra were recorded in the 190–700 nm range. For MS and MS2 detection, the dual AJS ESI source operated in negative ion mode. The gas temperature was set at 300 °C with a flow of 5 L/min, while the sheath gas temperature was 250 °C with a flow of 11 L/min. The nebulizer pressure was set at 35 psi, and the capillary and fragmentor voltages were 3500 V and 260 V, respectively. In the case of MS2 analysis, the compound fragmentation patterns were recorded at different collision energies (0, 10, and 20 eV) with an isolation width of 4 *m*/*z* [27]. The MassHunter Workstation Data Acquisition 10.0 (Agilent Technologies Inc., Santa Clara, CA, USA) program was used for data acquisition, while the MassHunter Qualitative Analysis 10.0 (Agilent Technologies Inc., Santa Clara, CA, USA) software was used for data processing.

### 3.3. Cell Viability and Antioxidant and Anti-Inflammatory Activities

#### 3.3.1. Cell Culture

The human epithelial cell line Caco-2 (ATCC^®^ HTB-37™, LGC Standards; Milan, Italy) was maintained in Eagle’s Minimum Essential Medium supplemented with 20% heat-inactivated fetal calf serum and 100 U/mL penicillin (all provided by Thermo Fisher Scientific; Milan, Italy). Cells were seeded at 4 × 10^4^ cells/mL and incubated at 37 °C, 5% CO_2_ in an air atmosphere. Twenty-four hours later, cell monolayers were gently washed, incubated in fresh culture media, and added with samples at concentrations ranging from 0 to 50% vol/vol.

#### 3.3.2. Cell Viability

To assess potential cytotoxic effects, we incubated Caco-2 cells for 24 h with the extracts at different concentrations. At the end of incubation, 5 mg/mL MTT (3-(4,5-dimethylthiazol-2-yl)-2,5-diphenyltetrazolium bromide, Merck; Milan, Italy) solution was added, and cells were incubated for additional 4 h at 37 °C. The resulting formazan crystals were then dissolved in 10% w/vol sodium dodecyl sulfate 10% containing 0.01N HCl. The absorbance was measured 16 h later at 590 nm using a microplate reader (MultiPlateReader VictorX2, Perkin Elmer; Milan, Italy).

#### 3.3.3. Detection of Intracellular Reactive Oxygen Species Levels

To detect intracellular reactive oxygen species (ROS), we used the cell-permeant 2′,7′-dichlorodihydrofluorescein diacetate (H_2_DCFDA) probe. Upon cleavage of the acetate groups by intracellular esterases and oxidation, the nonfluorescent H_2_DCFDA was converted to the highly fluorescent 2′,7′-dichlorofluorescein (DCF). The generation of DCF was directly proportional to cytoplasmic ROS. Caco-2 cells were incubated with extracts for 16 h. At the 15th hour of incubation, stimuli were renewed. Thirty minutes later, cells were loaded in warm PBS with 50 μM H_2_DCFDA (Molecular Probes, Invitrogen; Milan, Italy). Cells were incubated in the dark for 30 min at 37 °C and challenged with 25 μM H_2_O_2_ (Merck) [63]. Ten minutes later, cells were washed, harvested using Trypsin-EDTA, and analyzed using a BD FACSCantoTM flow cytometry system, collecting ten thousand events. Results were analyzed using Floreada.io software (https://floreada.io/analysis; accessed on 2 December 2024).

#### 3.3.4. Enzyme-Linked Immunosorbent Assay

Caco-2 cells were incubated for 16 h with extracts in the presence or not of 100 ng/mL lipopolysaccharide (LPS) from *Salmonella enterica* serotype *Typhimurium* (Merck). At the end of incubation, conditioned media were collected. Levels of interleukin (IL)-1β and tumour necrosis factor (TNF)-α were measured in the conditioned media using commercially available enzyme-linked immunosorbent assay kits (ELISA, ThermoFisher Scientific). ELISA kits are reported to have analytical sensitivity of 2 pg/mL and 0.13 pg/mL, respectively. Assays were developed using 3,3,5,5-tetramethylbenzidine (TMB), and data were recorded at 450 nm using a microplate reader (MultiPlateReader VictorX2, Perkin Elmer).

#### 3.3.5. Statistical Analysis

Chemical results were reported as mean ± standard deviation (SD) of two independent experiments, each one performed in duplicate, and biological results as mean ± standard error of the mean (SEM) of three independent experiments, each one performed in triplicate. Statistical analysis was performed using one-way analysis of variance (ANOVA) using GraphPad Prism v 7.05 (San Diego, CA, USA). Statistical significance was calculated using one-way ANOVA with Bonferroni multicomparison post-test. *p* values < 0.05 were considered statistically significant. In the chemical investigations, different Latin letters (which denote significant differences at *p* < 0.05) are attributed in alphabetic order from the richest samples to the poorest samples.

## 4. Conclusions

This study arises from the interest in assessing the potential application to human health of plant cell cultures from P. cerasus var. Marasca, a widely used variety for the production of food derivatives, which enriches the biodiversity of the Euganean Hills territory. The obtained undifferentiated in vitro cultures were characterized by high biomass production, interesting phytochemical profiles, and antioxidant and anti-inflammatory activity. Based on the results, we have confidence in a possible use of P. cerasus var. Marasca cell cultures as a sustainable alternative for the use of plant resources. Future research will focus on evaluating suspension culture conditions during the growth cycle, in order both to optimize the biomass production and to achieve a good secondary metabolite production, functional to the scale-up. The application of chemical and physical stresses will be evaluated as a strategy to increase the production of secondary metabolites or to trigger the production of secondary metabolites that are normally not produced.

## Figures and Tables

**Figure 1 molecules-30-01089-f001:**
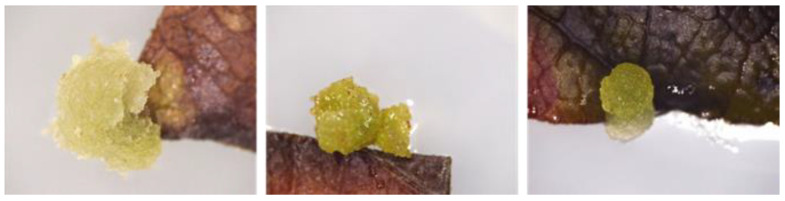
Magnifications of callogenesis starting at the cut ends of the explants.

**Figure 2 molecules-30-01089-f002:**
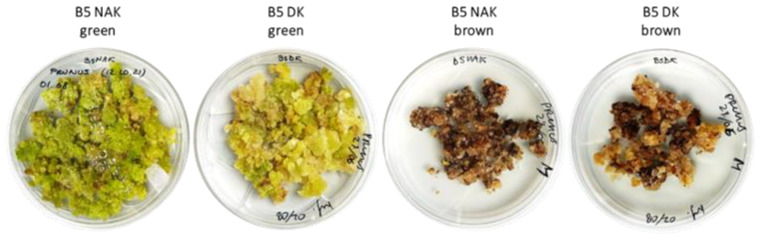
*Prunus cerasus* calli. Green line in B5 NAK, greenish line in B5 DK, brown line in B5 NAK, brown line in B5 DK.

**Figure 3 molecules-30-01089-f003:**
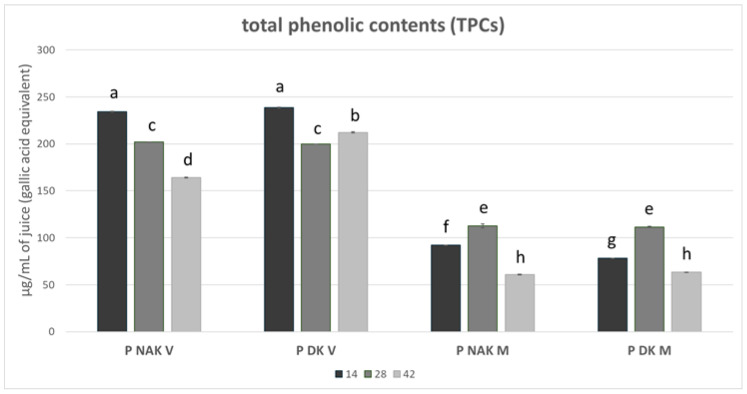
TPC of P NAK V, P DK V, P NAK M, P DK M juices during the growth cycle. The significant differences at *p* < 0.05 are denoted by different Latin letters.

**Figure 4 molecules-30-01089-f004:**
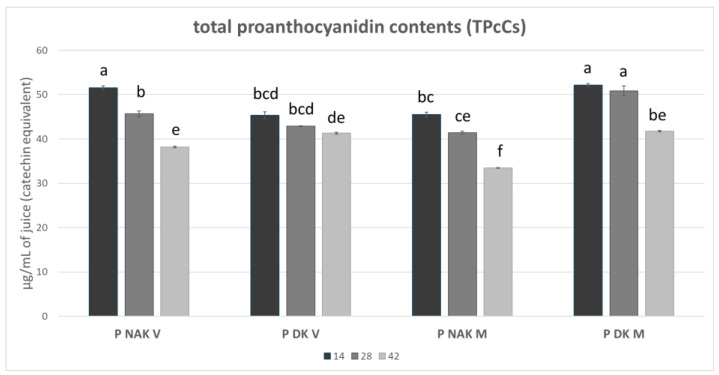
TPcC of P NAK V, P DK V, P NAK M, P DK M juices during the growth cycle. The significant differences at *p* < 0.05 are denoted by different Latin letters.

**Figure 5 molecules-30-01089-f005:**
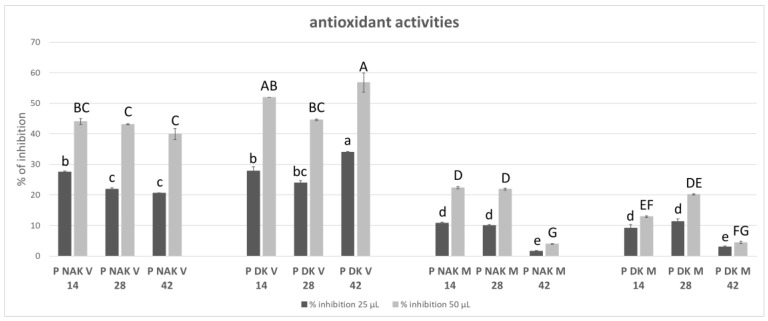
Percentage of DPPH inhibition of P NAK V, P DK V, P NAK M, P DK M juices during the growth cycle. For each time point, 25 and 50 µL of juices were tested for all the cell lines. The significant differences at *p* < 0.05 are denoted by different Latin letters: capital letters for the correlation among 50 µL of juices, and lowercase letters for the correlation among 25 µL of juices.

**Figure 6 molecules-30-01089-f006:**
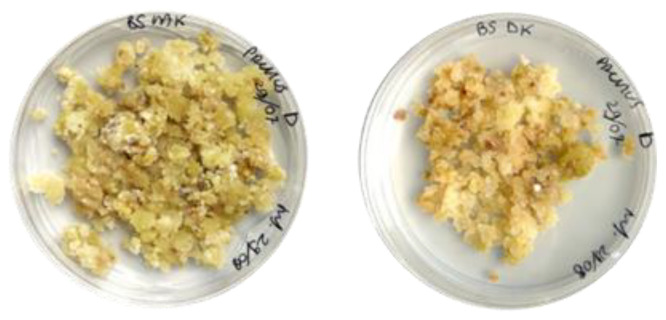
*Prunus cerasus* P NAK Vd (**left**) and P DK Vd (**right**).

**Figure 7 molecules-30-01089-f007:**
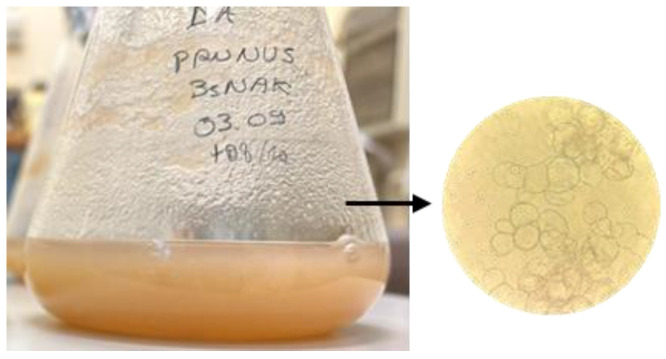
Suspension of P NAK V kept in darkness and magnification of cells (400×).

**Figure 8 molecules-30-01089-f008:**
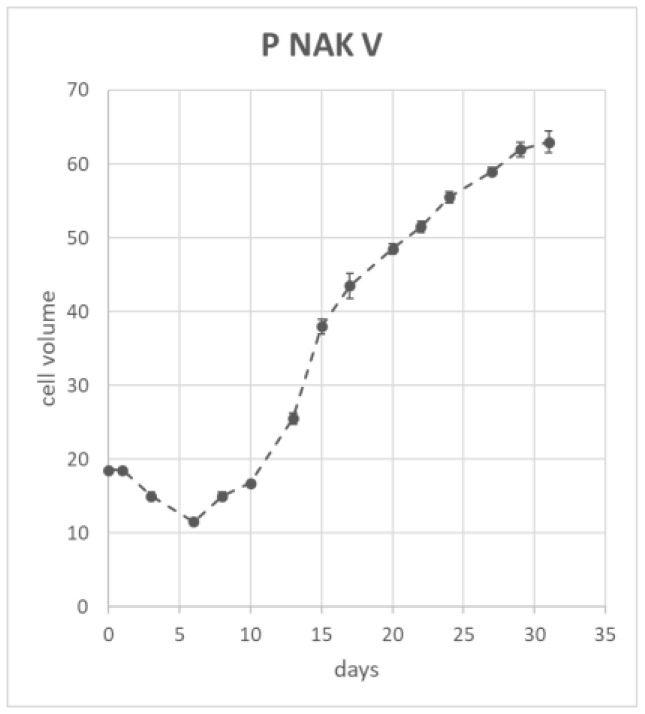
Growth curve of P NAK Vs.

**Figure 9 molecules-30-01089-f009:**
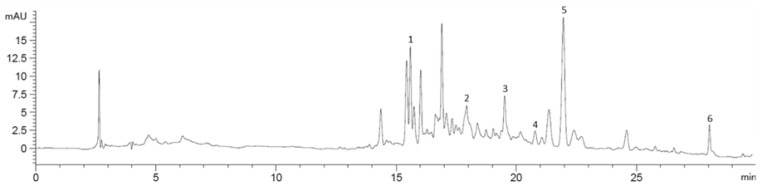
Chromatogram of P NAK V juice, acquired at 325 nm, with the identified peaks numbered.

**Figure 10 molecules-30-01089-f010:**
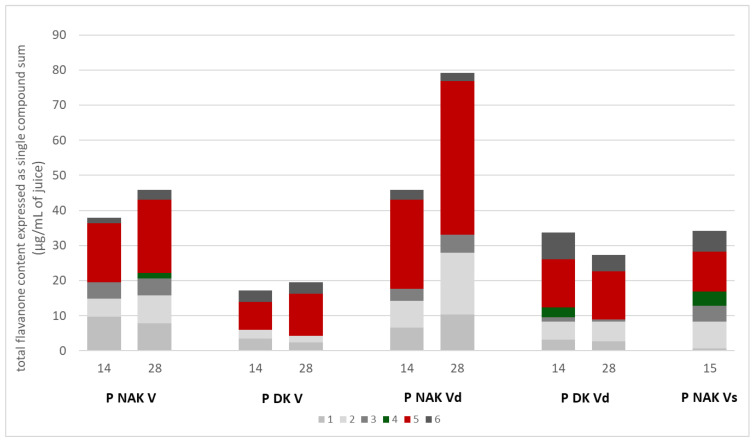
Flavanone derivative content of P NAK V, P DK V, P NAK Vd, P DK Vd, and P NAK Vs.

**Figure 11 molecules-30-01089-f011:**
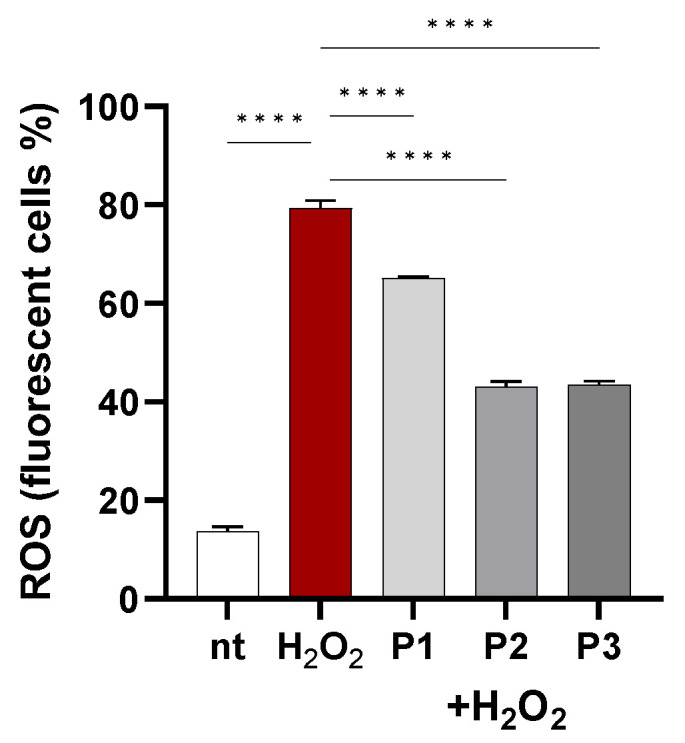
Caco-2 cells were incubated with extracts at the concentrations reported in Table 4 and then loaded with H_2_DCFDA. Intracellular ROS were detected by flow cytometry in 10^4^ cells. Data are reported as the percentage of fluorescent positive cells expressed as mean ± SEM of three experiments. **** indicates *p* < 0.0001. nt: non-treated cells.

**Figure 12 molecules-30-01089-f012:**
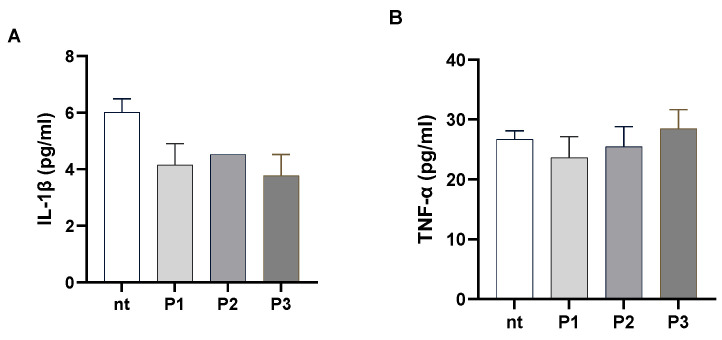
Caco-2 cells were treated with extracts at concentrations reported in Table 4 and incubated for 16 h. IL-1β (**A**) and TNF-α (**B**) levels were determined by ELISA in the conditioned media. Caco-2 cells were incubated with LPS 100 ng/mL and extracts for 16 h. IL-1β (**C**) and TNF-α (**D**) levels were determined by ELISA in the conditioned media. Data are reported as mean ± SEM of three experiments. **** indicates *p* < 0.02; * indicates *p* < 0.05 vs. LPS treated cells. nt: non-treated cells.

**Table 1 molecules-30-01089-t001:** Percentage of callogenesis and callus size for each medium tested.

Medium	Callogenesis (%)	Callus Size (mm)
B5 NAK	35	11–15
WPM NAK	64	1–3
MS NAK	-	-
B5 DK	43	4–7
WPM DK	10	8–10
MS DK	-	-

**Table 2 molecules-30-01089-t002:** Compounds tentatively identified by UHPLC-ESI-MS and HPLC-DAD analyses.

	RT UPLC-MS	RT HPLC-DAD	UV Max	[M-H]^−^ *m*/*z*	Fragmentation	TENTATIVE COMPOUND	P NAK V	P DK V	P NAK Vd	P DK Vd	P NAK Vs
1	19.4	15.4	218-280	433	293	flavanone derivative	X	X	X	X	X
2	21.1	16.9	220-284	433	277	flavanone derivative	X	X	X	X	X
3	25.2	19.5	222-284	433	287-206	flavanone derivative	X	-	X	X	X
4	27.0	20.7	228-288	413	367-271	naringenin derivative	X	-	-	X	X
5	28.7	22.0	224-282	433	271	naringenin glucoside	X	X	X	X	X
6	34.8	28.0	212-226-288	271		naringenin	X	X	X	X	X

**Table 3 molecules-30-01089-t003:** Total flavanone content of P NAK V, P DK V, P NAK Vd, P DK Vd, and P NAK Vs, expressed as µg/mL of juice. The significant differences at *p* < 0.05 are denoted by different Latin letters.

Cell Line	Growth Cycle Days	Total Flavanones (µg/mL of Juice)
P NAK V	14	37.92 ± 1.15 ^c^
	28	45.83 ± 1.31 ^b^
P DK V	14	17.22 ± 0.87 ^e^
	28	19.51 ± 1.43 ^e^
P NAK Vd	14	45.89 ± 0.95 ^b^
	28	79.22 ± 1.18 ^a^
P DK Vd	14	33.7 ± 1.25 ^c^
	28	27.32 ± 1.11 ^d^
P NAK Vs	15	34.24 ± 1.12 ^c^

**Table 4 molecules-30-01089-t004:** Cell viability was determined using MTT test in Caco-2 cells incubated with extracts for 24 h. The table reports the highest non-toxic concentrations evaluated in three experiments and the concentrations we chose to use in the subsequent experiments.

Samples	Highest Non-Toxic Concentration (% vol/vol)	Concentration Used (% vol/vol)
P 1	25	25
P 2	25	25
P 3	50	25

**Table 5 molecules-30-01089-t005:** Basal media and hormone balances. 2,4-D (2,4-dichlorophenoxyacetic acid); NAA (α-naphtaleneacetic acid); K (kinetin).

Medium	Basal Medium	2,4-D mg/L	NAA mg/L	K mg/L
B5-NAK	B5		1	1
WPM-NAK	WPM		1	1
MS-NAK	MS		1	1
B5-DK	B5	1		1
WPM-DK	WPM	1		1
MS-DK	MS	1		1

## Data Availability

The presented data are available on request from the authors.

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
