# Peer review of "From “Maraschino” to Cell Cultures: A Deep Study on Prunus cerasus L. Cell Culture Juices"

_molecules, 2025, doi:10.3390/molecules30051089_

Round 1
Reviewer 1 Report
Comments and Suggestions for Authors
The manuscript demonstrates the in vitro cultivation of cell cultures of P. cerasus var. Marasca. The manuscript presents the challenging aspects of obtaining in vitro cultures, with particular emphasis on the process of medium selection and sterilization.
This paper is potentially interesting and contain a practical information regarding the methodology for initiating the in vitro cultures, but there are some issues.
- - The poor quality of Figures 3,4 and 8
- - The identification of interesting phytochemical profiles (mentioned in the conclusion) is based on detection of the content of total phenols and total proanthocyanidins. However, using the MS method identified only flavones; why the other target compounds, proanthocyanidins, were not identified (theoretically their oligomers could be identified by MS)?
Author Response
Dear Reviewers,
here is the manuscript “From "Maraschino" to cell cultures: in a deep study on Prunus cerasus L. cell culture juices” revised according to Your suggestions.
First of all, we want to thank You for the comments, as they helped us to improve our manuscript's quality.
All the changes are outlined in the manuscript in red.
Since You asked to write the manuscript in more corrected English, we deeply reconsidered a large part of the manuscript.
Reviewer 1:
We improved the quality of Figures 3,4,8 together with 5 and 10 for better visualization.
About the comment on why we did not identify proanthocyanidins, we really appreciated having the opportunity to clarify this point with you. We performed commonly-used preliminary colorimetric assays to have an idea of the chemical fingerprint of our materials in order to select the best ones for going on. We decided to investigate the total phenol content because this method “can be considered to be the most widely used in laboratories around the world, to quantify the total polyphenols content” as reported by (Talanta 272 (2024) 125771); therefore, it can be considered an easy method to compare different materials and to have an idea of the phenols inside. Moreover, we investigated the total proanthocyanidin content because, as reported in the text, this compound class is receiving huge attention for its possible utilization in the nutraceutical field. Following this initial screening, we chose the most promising cell lines, and of these, we analyzed the juices. You are absolutely right that proanthocyanidins could be detected in LC-MS analyses, but after the initial screening, seeing the significant abundance of flavanone derivatives by LC-MS-DAD, we decided to focus our qualitative and quantitative analyses on these secondary metabolites, which are in any case themselves only a part of the total phenol content. In addition, as it is written in the paper, we pursued the flavanone derivative identification and quantification because the naringenin was, as far as now, detected only in heartwood, pollen, and kernel oil, thus, in our opinion, the quantification of it and its derivatives could represent a new insight into the cell totipotency of plant cell cultures.
Reviewer 2:
We changed the list of the keywords as You suggested. Furthermore, we changed the Abstract and the Introduction stressing the aim of the work.
We rewrote a large part of the manuscript in order to improve the English form, and we made the Manuscript easier to read. In the new version, many confusing parts are eliminated or explained better. We added references and literature data for easy comparison, and we moved into the “Materials and methods” some paragraphs of “Results and discussion” section and vice versa. We improved the Figures.
Reviewer 3:
As the Reviewer suggested, we changed the Abstract and the Introduction, stressing our scope better, and improved the figures. We explained the abbreviations when used for the first time. All the analyses were described with a statistic significance of our data.
For what concerns the quantitative calculation of naringenin derivatives, we are perfectly conscious that it could be better to perform the analysis comparing the juice metabolites with their reference standards. However, considering the similarity of the UV spectra of the isolated flavanones, we think that a quantitative analysis of a compound mixture (belonging to the same class and showing the same UV spectrum characteristics) by single marker could be accurate enough.
Best regards,
Vanessa Dalla Costa

Reviewer 2 Report
Comments and Suggestions for Authors
The present work provides information on the chemical profile and antioxidant/anti-inflammatory activity of juices obtained from Prunus cerasus var. Marasca calli and cell cultures grown in vitro. However, the manuscript has several serious weaknesses and, in my opinion, is not acceptable for publication in its current form. Accordingly, I would like to suggest to the authors to completely rewrite the manuscript according to the comments. I believe that this would facilitate the interpretation of the results and would certainly improve the quality of the manuscript.
Some comments and suggestions to the authors are listed below:
The list of keywords should be shortened. Please choose 5 to a maximum 7 keywords that best describe your work. For example: Prunus cerasus; plant calli; suspension cultures; naringenin; anti-inflammatory activity; antioxidant activity. In addition, the main objective of the work should be written more precisely and clearly.
The Results and discussion section is generally confusing and very poorly written, and this is in my opinion, the major weakness of the manuscript. Although a lot of experimental data are presented, most of them are only described but hardly or not discussed at all. To improve the manuscript, the authors should organize their experimental results well and compare them with some previous studies and provide an overview of the current literature on this topic.
Parts of the text from the subsection 2.1. In vitro culture establishment, preliminary colorimetric assays and cell line selection, especially subsection 2.1.1. (Callus culture establishment) should be transferred to the Materials and methods section (for example, text between lines 114-120).
Lines 135-145: How exactly did the authors select type/concentrations/combinations of plant growth regulators used for callus culture establishment of? In was stated: “based on literature data on the genus or species under consideration”, but there are no references in the text to support this. Please explain this in detail.
Lines 146-148: The sentence “Callus formation initiated in correspondence with the cutting side of the explants” is quite confusing. Please rewrite it.
Lines 176-177: How do the authors come to the conclusion that the “growth rate of the brown cell lines was extremely lower than of the green ones”? Is this just based on observation or is there data to support this statement? Please explain.
Lines 191-197: This part of the text is quite confusing and unclear. Please rewrite it.
Lines 245-246: The authors state that “The overall trend of the antioxidant activity is in agreement with the total phenol content”. Please explain. Did the authors mean that antioxidant activity is correlated with TPC? If so, this assertion should be supported by the experimental data.
In addition, the results shown in Figure 5 are quite confusing. What exactly does “%inhibiz 25” or “%inhibiz 50” mean? I would suggest that the authors change the legends of each graph to make them clearer.
Line 250: What is the “richest sample”? Please explain.
Lines 288-321: Subsection 2.2. Suspension cultures should be rewritten; it is too broad and descriptive and contains parts that belong in the Materials and methods section.
Lines 326-329: The authors stated: "'Due to the low flux during the UHPLC analysis, leading to a different compound elution, the retention times of the compounds identified by MS, were compared to the ones obtained by HPLC-DAD, used below to carry out the quantitative analyses." What exactly did the authors do here? What is the low flux during analysis? How could the retention times of the compounds detected with two different systems/methods/analyses be compared? Please explain this in detail.
Lines 460-461: The authors wrote: "The concentrations of juices used for the anti-inflammatory assays are reported in Table x, and are...". Did they mean Table 4? Please add the missing data.
Lines 465-466: It is written: "Incubation with LPS enhanced the production of IL-1β and TNF-α compared with not treated cells (Figure 3, p<0.0001)." Which table exactly are they referring to? Are the two graphs in Line 472 also part of Figure 12 or not? If the latter is the case, a corresponding figure legend should be added.
Lines 501-502: What type of liquid medium was used for the suspension cultures? Please specify.
Subsection 4.2.1. Juice preparation is very poorly written. What did the author mean by "the whole procedure has been standardised in terms of quantity and timing". What is "the same consistency" of the plant material? Please explain the extract preparation process in detail.
Lines 534-538: This part of the text is quite confusing. Were the samples analyzed in triplicate or in duplicate? Please revise it.
Comments on the Quality of English LanguageThe English could be improved to more clearly express the research.
Author Response

(The authors gave the same response as above.)

Reviewer 3 Report
Comments and Suggestions for Authors
The subject of the article falls within the scope of Molecules. The authors derived in vitro cultures of P. cerasus var. marasca, selected the most effective cell lines in terms of secondary metabolite production and biological activity. I appreciate the work the authors put into the research, but in its current form the article cannot be published. I ask the authors to recompose the article according to the suggestions:
The Abstract is too general, it should present the main results obtained.
The Introduction lacks an explanation of whether there are publications on in vitro cultures of the studied species or genus Cerasus and what is innovative in the research currently conducted by the authors. The authors should emphasize what has been done for the first time. The content of the sentence in lines 92-93 is unclear.
When discussing the results, please change the style. The authors use a lot of very general formulations, e.g. in lines 98-99; 103-106; 107-113; 130-134; 264-265
I do not understand the frequent emphasis that some part of the work is time consuming, e.g. on line 163; this is obvious for all studies on in vitro cultures.
Please explain the abbreviations when they are used for the first time.
On lines 176-7 the authors formulated the conclusion "Furthermore, the growth rate of the brown cell lines was extremely lower than of the green ones." - there is no table or figure with the results on the basis of which the authors conclude so.
Figures 3, 4, 5 are illegible.
In how many repetitions were the analyses performed? Why are the results not statistically developed? All conclusions without prior determination of statistically significant differences are unfounded. I ask the authors to supplement the statistics in all analyses.
The term “hormones” is reserved for naturally occurring substances. In reference to, for example, 2,4-D, the term “plant growth regulators” should be used.
Lines 393-395 the use of the phrase “totipotency” does not fit the context.
How were quantitative calculations of naringenin derivatives made? As I understand from the methodology, the authors had only one commercial standard (naringenin), so only for this compound could quantitative calculations be made. The remaining compounds could only be qualitatively confirmed by MS/MS.
The Conclusions part, should indicate further research directions.
Comments on the Quality of English Language
Please proofread the text because some of the sentences are unclear.
Author Response

(The authors gave the same response as above.)

Round 2
Reviewer 2 Report
Comments and Suggestions for Authors
The manuscript is now much improved and can, in my opinion, be accepted for publication after minor textual corrections.
Line 274: Please, write “capital letters” correctly
Lines 367-377: Table 2 and Figure 9 should be rearranged according to the text
Table 2, row 4: Please, write “222-284” instead of “sh222-284”
Comments on the Quality of English LanguageThe English could be improved
Reviewer 3 Report
Comments and Suggestions for Authors
The authors have made the suggested corrections and clarified the doubts. The article can be published in this form.